# Dual-Energy CT-Based Bone Mineral Density Has Practical Value for Osteoporosis Screening around the Knee

**DOI:** 10.3390/medicina58081085

**Published:** 2022-08-11

**Authors:** Keun Young Choi, Sheen-Woo Lee, Yong In, Man Soo Kim, Yong Deok Kim, Seung-yeol Lee, Jin-Woo Lee, In Jun Koh

**Affiliations:** 1Joint Replacement Center, Eunpyeong St. Mary’s Hospital, Seoul 03312, Korea; 2Department of Orthopaedic Surgery, College of Medicine, The Catholic University of Korea, Seoul 06591, Korea; 3Department of Radiology, Eunpyeong St. Mary’s Hospital, Seoul 03312, Korea; 4Department of Orthopaedic Surgery, Seoul St. Mary’s Hospital, Seoul 06591, Korea

**Keywords:** dual-energy CT, Hounsfield unit, bone mineral density, volumetric phantomless BMD, opportunistic CT

## Abstract

*Introduction*: Adequate bone quality is essential for long term biologic fixation of cementless total knee arthroplasty (TKA). Recently, vertebral bone quality evaluation using dual-energy computed tomography (DECT) has been introduced. However, the DECT bone mineral density (BMD) in peripheral skeleton has not been correlated with Hounsfield units (HU) or central dual-energy X-ray absorptiometry (DXA), and the accuracy remains unclear. *Materials and methods*: Medical records of 117 patients who underwent TKA were reviewed. DXA was completed within three months before surgery. DECT was performed with third-generation dual source CT in dual-energy mode. Correlations between DXA, DECT BMD and HU for central and periarticular regions were analyzed. Receiver operating characteristic (ROC) curves were plotted and area under the curve (AUC), optimal threshold, and sensitivity and specificity of each region of interest (ROI) were calculated. *Results:* Central DXA BMD was correlated with DECT BMD and HU in ROIs both centrally and around the knee (all *p* < 0.01). The diagnostic accuracy of DECT BMD was higher than that of DECT HU and was also higher when the T-score for second lumbar vertebra (L2), rather than for the femur neck, was used as the reference standard (all AUC values: L2 > femur neck; DECT BMD > DECT HU, respectively). Using the DXA T-score at L2 as the reference standard, the optimal DECT BMD cut-off values for osteoporosis were 89.2 mg/cm^3^ in the distal femur and 78.3 mg/cm^3^ in the proximal tibia. *Conclusion*: Opportunistic volumetric BMD assessment using DECT is accurate and relatively simple, and does not require extra equipment. DECT BMD and HU are useful for osteoporosis screening before cementless TKA.

## 1. Introduction

The number of younger and more active patients treated with total knee arthroplasty (TKA) continues to increase rapidly [1,2,3,4,5], but TKA in younger patients has exhibited a lower 10-year survivorship with higher demand for revision [6,7,8,9]. Thus, long-term and biologic fixation after TKA has been revisited, and interest in cementless TKA, which theoretically enables physiologic fixation, has increased [10,11]. The earlier cementless TKA prostheses had unacceptably high failure rates and poor clinical outcomes [12,13], but major advances in design and materials in recent years have greatly improved the newer generation of cementless prostheses [14,15,16]. Stereo imaging analysis has shown that the new cementless prosthesis is superior to traditional press fit design and cemented prostheses in reducing micromotion, and it is anticipated that the risk of aseptic loosening will be greatly reduced [17].

Preoperative bone quality and bone mineral density (BMD) around the knee joint are strongly associated with adequate bone ingrowth and initial prosthesis stability after cementless TKA [16,18,19]. However, there is no gold standard method as yet to evaluate BMD around the knee joint. Previous study has shown that BMD around the knee can be estimated by CT attenuation in Hounsfield units (HU), which are a measure of the standardized linear attenuation coefficient of CT [20]. However, the HU-based BMD is limited due to distortion by changes in marrow composition, and diagnostic value around the knee joint remains unclear [20,21,22,23]. Dual-energy CT (DECT), which acquires CT attenuation data at two different energy levels, can generate images of soft tissue or bone marrow by decomposing different tissue characteristics [24]. It has recently been proposed as a tool to screen for osteoporosis with greater diagnostic accuracy than traditional quantitative CT (QCT) [22,25]. However, the DECT BMD for peripheral bone assessment has not been studied yet.

Therefore, the purposes of this study were (1) to assess the strength of correlation between central dual-energy X-ray absorptiometry (DXA) BMD, DECT BMD and HU at the lumbar spine, femur neck, distal femur, and proximal tibia; and (2) to calculate the diagnostic accuracy of DECT BMD and HU for bone quality assessment around the knee joint when the T-score of central DXA is used as the reference standard. We hypothesized that the volumetric BMD and HU from DECT would be well correlated with DXA BMD. We also hypothesized that the diagnostic accuracy of DECT BMD and HU in bone quality assessment around the knee joint would be high.

## 2. Material and Methods

A total of 125 patients who underwent TKA between November 2021 and January 2022 and had both preoperative third-generation dual-source DECT and DXA within three months before surgery were considered for study inclusion. The study was approved by the Institutional Review Board of our institute (PC22RISI0049) and it was exempted from informed consent because it is a retrospective medical and radiological record review. Patients with metal prosthesis, fracture or infection in central (spine or hip) or peripheral (distal femur or proximal tibia) regions were excluded (Figure 1). DXA at the lumbar spine and left pelvic area had been taken within 3 months before TKA (Figure 2A), and had identified osteoporosis in 11% of L2 vertebral bodies and in 14.5% of femur necks (Table 1). Lower extremity CT was performed for the purpose of preoperative planning for TKA with a measured resection technique. All CT studies were performed on a third-generation dual-source CT system in dual-energy mode (SOMATOM Force; Siemens Healthineers, Erlangen, Germany), with tube A at 90 kVp and 180 mAs and tube B at Sn150 kVp [0.64-mm tin filter] and 180 mAs. Image series were collected in a craniocaudal direction with the patient in a supine position without administration of a contrast agent. Three image sets, 90 kVp, Sn150 kVp, and weighted average (ratio 0.5:0.5), were acquired in each CT examination to resemble the contrast properties of single-energy bone CT images. The images were reconstructed with a dual-energy bone kernel (B69f), transferred to the image archiving system (SyngoVia, Siemens Healthineers, Erlangen, Germany), and then transferred to a personal computer carrying the analysis software for postprocessing.

Phantomless volumetric BMD assessment of L2 and the left femur neck with DECT requires manual delineation of trabecular volumes of interest (VOI) in the L2 vertebra and the left pelvis, which was carried out by one of the authors with dedicated software (Examine, Siemens Healthcare, Erlangen, Germany). Regions of interest (ROIs) defined to best include trabecular bone and exclude any cortical bone were also drawn manually on the images loaded into the software (Figure 2B). The software then performed calculations according to a dedicated algorithm and the resulting output included the volumetric BMD values.

For DECT HU analysis, one of the authors, working at a conventional PACS workstation, manually defined polygonal ROIs on standard bone reconstructions in sagittal or axial image series (Figure 2C). The ROIs were positioned in the anterior trabecular bone space of the L2 vertebral body, as proposed by several studies [26,27,28]. Thereby, the reader was instructed to avoid attenuation heterogeneity by placing the ROIs in areas of spinal hemangiomas or other causes of attenuation heterogeneity. All HU values were obtained as averages of three serial polygonal ROIs.

Correlation between DXA BMD or T-scores for L2 or the left femur neck and DECT BMD or HU of L2, left femur neck, distal femur, and proximal tibia was analyzed and receiver-operating characteristic (ROC) curves were plotted to evaluate diagnostic accuracy. In addition, the area under the curve (AUC), optimal threshold, and sensitivity and specificity of DECT BMD and HU for distal femur and proximal tibia were determined by using the DXA T-scores for L2 and femur neck as the reference standards.

The values assessed by the researchers were obtained as the average of two evaluators. Each researcher assessed every radiological variable two times with the interval at least two weeks. Intra- and inter-observer reliability for each measurement were expressed as intraclass correlation coefficients (ICCs).

### Statistical Analysis

All computations were performed with Statistical Package for Social Sciences (SPSS) version 21 (IBM Corp., Armonk, NY, USA), with significance set at *p* < 0.05. Variables are presented as mean ± standard deviation. The correlation between DXA BMD and DECT BMD or HU was analyzed by using the Pearson product moment correlation. ROC curve analysis and calculation of the AUC were performed to evaluate optimal cut-off values for distinguishing osteoporosis from normal BMD. Osteoporosis [T < −2.5], osteopenia [−1.0 ≤ T ≤ −2.5], and normal BMD [T > −1.0] were classified according to the DXA T-score. Sensitivity, specificity, positive and negative predictive values (PPV and NPV), and accuracy were computed from these cut-off values.

## 3. Results

Of a total of 125 patients, three patients with a history of metal implants in spine or hip, tumor or previous fracture at spine or hip, previous vertebroplasty or kyphoplasty at spine, or previous infection at spine or hip were excluded. Five patients with metal implant, previous fracture, or history of previous osteotomy around a knee were excluded. Thus, a total of 117 patients were finally included and their medical and radiographic data were reviewed (Figure 1). The mean age was 70.6 years and the mean body mass index (BMI) was 26.6 kg/m^2^ (Table 1).

DXA BMD of L2 and femur neck were strongly correlated with DECT BMD and HU at their own region (all *r* > 0.5, all *p* < 0.01). In addition, the DXA BMD of L2 and the femur neck were significantly correlated with DECT BMD and HU in the other ROIs, surpassing moderate degree (all *r >* 0.3), except for the DXA BMD of the femur neck and HU of the proximal tibia (*r =* 0.286) (all *p* < 0.01) (Table 2). In addition, the correlation value (*r*) was higher for L2 than for the femur neck for all ROIs except for their own region.

The DECT BMD showed a stratified result from normal bone quality for osteoporosis measured by the DXA T-score in every ROI (Figure 3A–D). When the DXA T-score at L2 was used as the reference standard, the optimal cut-off values of DECT BMD for diagnosing osteoporosis were calculated as 89.2 mg/cm^3^ in the distal femur and as 78.3 mg/cm^3^ in the proximal tibia. In terms of DECT HU, the optimal cut-off values were 104.5 in the distal femur and 66.5 in the proximal tibia. When the DXA T-score at the left femur neck was used as the reference standard, the optimal DECT BMD cut-off values for diagnosing osteoporosis were 96.9 mg/cm^3^ in the distal femur and 80.9 mg/cm^3^ in the proximal tibia and the optimal cut-off values of DECT HU were 117.4 in the distal femur and 66.8 in the proximal tibia (Table 3). In addition, all AUC values of L2 surpassed femur neck and AUC values of DECT BMD surpassed DECT HU, respectively (Table 3). Thus, the diagnostic accuracy of both DECT BMD and HU was better when the DXA T-score for L2, rather than the femur neck, was used as the reference standard (Figure 4A,B).

Intra- and inter-observer reliability for all radiographic measurements was considered acceptable, ranging from 0.81 to 0.99 and 0.81 to 0.96, respectively.

## 4. Discussion

Aseptic loosening is a major cause of failure after TKA in young and active patients, and the need for optimal biological fixation of cementless prostheses is growing [1,6,7,8,9,29,30]. Preoperative poor bone quality is associated with a higher rate of micromotion and migration in cementless prostheses. Meanwhile, there is no gold standard method for evaluating bone quality in peripheral regions [19,31]. We investigated the diagnostic accuracy of DECT BMD and HU for osteoporosis around the knee and compared the DECT values with the central DXA. In addition, we calculated the optimal cut-off values of DECT BMD and HU for diagnosing osteoporosis around the knee.

We found that BMD of central DXA was significantly correlated with DECT BMD and HU in both central and peripheral ROIs. The DXA BMD and T-scores for L2 and the femur neck were strongly correlated with DECT BMD and HU in their own region (all correlation coefficients > 0.5). In addition, the DXA BMD of L2 and the femur neck were moderately correlated with DECT BMD and HU in all other ROIs, except for a weak correlation of the DXA BMD at the femur neck with DECT HU at the proximal tibia. Our results concur with previous studies reporting the correlation between DXA BMD and radiologic values obtained from opportunistic CT scans [20,26,27]. However, while a substantial number of studies have found correlations between central DXA BMD and DECT BMD or HU [20,21,22,26,27], it is difficult to compare those studies with ours because all of the previous studies were confined to ROIs other than the knee joint. According to our study, DECT BMD and HU were more strongly correlated with DXA BMD at L2 than at the femur neck. In addition, there was stronger correlation between DECT BMD than DECT HU with DXA BMD. These results concur with a previous study reporting a stronger correlation between DXA BMD and DECT BMD than DECT HU [21]. Our study results demonstrate that DECT can yield more accurate and precise volumetric BMD than HU, permitting opportunistic BMD measurements in routine CT scans without the need for calibration phantoms. Our results indicate that DECT for the routine preoperative evaluation of lower extremity axial alignment can also be a useful method of evaluating preoperative bone quality in patients who are about to undergo cementless TKA.

The result of this study, using DXA T-score as the reference standard, also supports the hypothesis that DECT BMD and HU can provide reliable diagnostic accuracy in assessing bone quality around the knee joint. DECT BMD provided the highest diagnostic accuracy with DXA BMD at L2 as the reference standard (all AUC values: L2 > femur neck; DECT BMD > DECT HU, respectively) (Table 3). Moreover, the reference of L2 provided more precise diagnostic value in the tibia than in the femur. This result concurs with previous studies finding that phantomless volumetric DECT BMD offered significantly more accurate BMD assessment and superior diagnostic accuracy for osteoporosis than DECT HU [21]. Our results suggest that DECT BMD has the potential to be a gold standard method for evaluation of BMD around the knee joint. However, because this finding is based on the central DXA T-score as the reference standard, further biomechanical research regarding the correlation between DECT BMD or HU and the true bone strength around the knee is required.

Interestingly, BMD evaluation using DECT may have additional benefits and possibilities compared to DXA. Although DXA has been regarded as a gold standard preferred by the International Society for Clinical Densitometry (ISCD), it has certain limitations in clinical application, and variations in body composition can lead to up to 20% error [32]. DXA cannot be used in patients with scoliosis or calcifications from chronic disease, metal implants in both hips or at multiple levels of the spine, or cement in a vertebral body. In addition, DXA is based in two dimensions, meaning it cannot distinguish cancellous bone from cortical bone quality. On the other hand, the three-dimensional DECT analysis can discriminate cancellous bone, which may comprise most of the bone-prosthesis interface between cementless TKA and cortical bone. There are already studies reporting a higher rate of successful fusion and lower rates of adjacent spine fracture following spine surgery in patients with higher HU [33,34,35,36]. Moreover, there also are studies reporting no significant difference between HU measurements on sagittal and axial CT images [27]. These previous studies may further support the possibility of utilizing DECT BMD and HU as a tool for evaluation of preoperative bone quality around the knee in patients undergoing TKA. This study assessing BMD around the knee joint proposes a novel method in evaluating bone quality around peripheral regions. Moreover, there is a possibility to expand this method to other peripheral joints. If surgeons can get information about a patient’s bone quality around the knee or other peripheral joints from a preoperative CT scan, they may be able to screen for the suitability of a cementless implant and modify their technique or type of implant according to the findings. Routine preoperative DECT may be helpful for detecting unexpected poor bone quality that could provoke early prosthesis failure.

This study had several limitations. First, it was confined to an Asian population, and most of the patients undergoing TKA who were included in this study are women (102/117, 87%), and it may be difficult to generalize our results to other ethnicities. It is not yet clear why the majority of patients with arthritis in Korea are females [37]. Second, the study was confined to middle-aged and older patients (70.6 ± 6.5, 54 to 88) because only patients who underwent TKA were included. Third, although there was significant correlation between DXA BMD and DECT BMD, the clinical meaning of DECT BMD values remains unclear. As a way to overcome this limitation, further study examining the direct correlation between DECT BMD values and the properties of actual bone is required. Fourth, there are concerns of increased radiation exposure by utilizing DECT. However, DECT has been reported not to increase radiation exposure compared with single energy CT [38]. In addition, preoperative lower extremity CT is already part of the routine preoperative planning protocol for TKA by measured resection technique. Thus, the opportunistic measurement of BMD around the knee using DECT does not seem to increase radiation exposure compared with standard clinical practice. Fifth, because of the small study population, it is possible that the study was underpowered and subject to type-II error with respect to detecting all relevant outcomes. Sixth, analysis on factors possibly affecting BMD is not done in this study. Further study focused on the possible factors is required to expand the clinical relevance and would provide more valuable information. Finally, there is a possibility of limited reproducibility because of discrepancies in HU and BMD values among institutions with different CT scanners, DXA devices, and software [20]. Nonetheless, it is still possible to use DECT BMD as a tool to evaluate BMD around the knee, and despite these limitations, our study appears to provide valuable information on the practical use of DECT for osteoporosis screening around the knee in patients scheduled for cementless TKA.

## 5. Conclusions

Volumetric BMD assessment using DECT is accurate, relatively simple, and does not require further equipment. DECT BMD and HU are useful tools for osteoporosis screening before cementless TKA in clinical practice. Further research focused on the correspondence between the actual bone quality of the distal femur and proximal tibia and DECT BMD and HU is required to expand the clinical relevance.

## Figures and Tables

**Figure 1 medicina-58-01085-f001:**
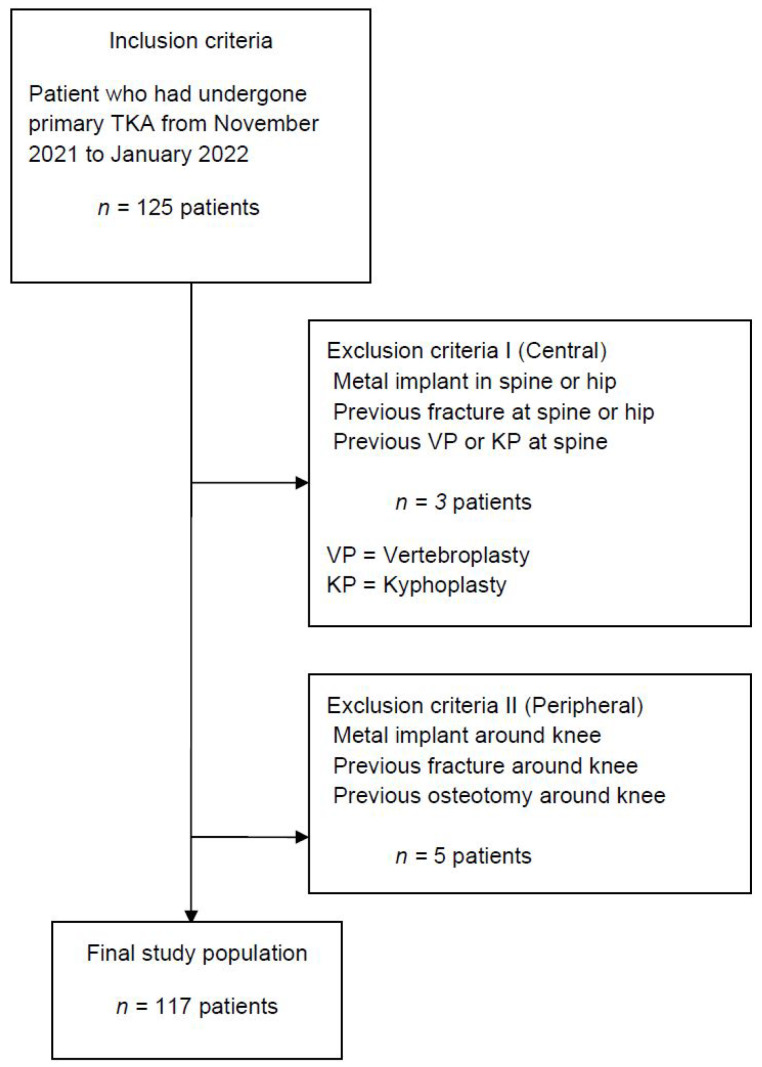
Standards for Reporting of Diagnostic Accuracy Studies (STARD) flow chart of patient inclusion.

**Figure 2 medicina-58-01085-f002:**
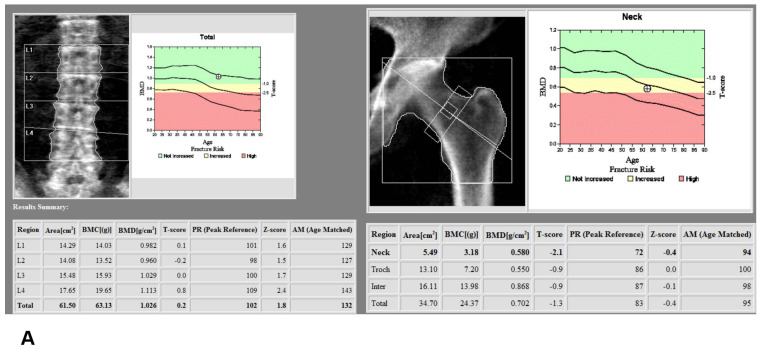
(**A**) Assessment of dual x-ray absorptiometry (DXA) in the lumbar spine and femur neck; (**B**) Manual definition of the region of interest (ROI) and assessment of bone mineral density (BMD) derived from dual-energy computed tomography (DECT) in the distal femur and proximal tibia using dedicated DECT postprocessing software; (**C**) Manual definition of the region of interest (ROI) and assessment of Hounsfield unit (HU) derived from dual-energy computed tomography (DECT) in the distal femur and proximal tibia using dedicated DECT postprocessing software.

**Figure 3 medicina-58-01085-f003:**
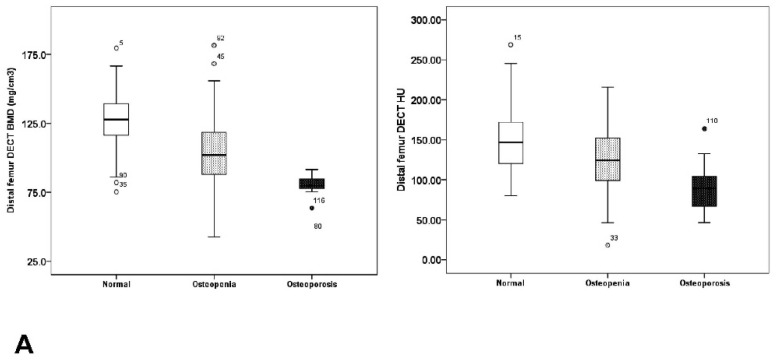
(**A**) Box plots of volumetric bone mineral density (BMD) and Hounsfield unit (HU) values in the distal femur derived from dual-energy computed tomography (DECT). The plots show the distribution of values that were categorized as normal BMD, osteopenia, and osteoporosis according to the dual x-ray absorptiometry (DXA)-derived T-score in the lumbar spine, which served as the reference standard; (**B**) Box plots of volumetric bone mineral density (BMD) and Hounsfield unit (HU) values in the proximal tibia derived from dual-energy computed tomography (DECT). The plots show the distribution of values that were categorized as normal BMD, osteopenia, and osteoporosis according to the dual x-ray absorptiometry (DXA)-derived T-score in the lumbar spine, which served as the reference standard; (**C**) Box plots of volumetric bone mineral density (BMD) and Hounsfield unit (HU) values in the distal femur derived from dual-energy computed tomography (DECT). The plots show the distribution of values that were categorized as normal BMD, osteopenia, and osteoporosis according to the dual x-ray absorptiometry (DXA)-derived T-score in the femur neck, which served as the reference standard; (**D**) Box plots of volumetric bone mineral density (BMD) and Hounsfield unit (HU) values in the proximal tibia derived from dual-energy computed tomography (DECT). The plots show the distribution of values that were categorized as normal BMD, osteopenia, and osteoporosis according to the dual x-ray absorptiometry (DXA)-derived T-score in the femur neck, which served as the reference standard.

**Figure 4 medicina-58-01085-f004:**
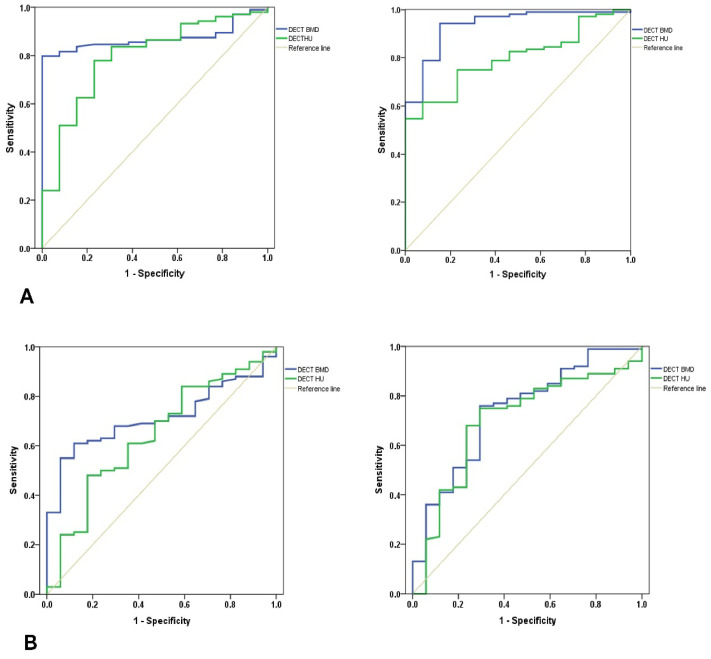
(**A**) Representative receiver operating characteristic (ROC) curves of phantomless volumetric bone mineral density (BMD) (blue line) values and Hounsfield unit (HU) measurements (green line) derived from dual-energy computed tomography (DECT) for the detection of osteoporosis using the dual x-ray absorptiometry (DXA) derived T-score of the lumbar spine as the reference standard. (Left: values in distal femur; Right: values in proximal tibia); (**B**) Representative receiver operating characteristic (ROC) curves of phantomless volumetric bone mineral density (BMD) (blue line) values and Hounsfield unit (HU) measurements (green line) derived from dual-energy computed tomography (DECT) for the detection of osteoporosis using the dual x-ray absorptiometry (DXA) derived T-score of the femur neck as the reference standard. (Left: values in distal femur; Right: values in proximal tibia).

**Table 1 medicina-58-01085-t001:** Patient demographics and preoperative characteristics.

Demographic Data	*n* = 117	
Age *	70.6 ± 6.5 (54~88)	
Sex (male: female) ^†^	15 (13): 102 (87)	
Height (cm) *	153.7 ± 7.2 (140~178)	
Weight (kg) *	63.0 ± 10.6 (44~89)	
BMI (kg/m^2^) *	26.6 ± 3.4 (21.0~35.2)	
Diagnosis of Osteoporosis (%) ^†^	L2	Femur neck
Normal	44 (38)	31 (26.5)
Osteopenia	60 (51)	69 (59)
Osteoporosis	13 (11)	17 (14.5)
DXA	L2	Femur neck
BMD *	0.859 ± 0.166 (0.556~1.468)	0.640 ± 0.097 (0.451~0.945)
T-score *	−1.055 ± 1.419 (−3.7~3.3)	−1.532 ± 0.899 (−3.3~1.3)

^†^ Data are presented as numbers (percentage) of patients. * Data are presented as the means ± standard deviations (range). DXA, dual x-ray absorptiometry; BMD, bone mineral density.

**Table 2 medicina-58-01085-t002:** Correlation analysis between DECT BMD or HU and central DXA BMD or T-score.

		DXA							
		L2				Femur Neck			
		BMD (g/cm^2^)		T-Score		BMD (g/cm^2^)		T-Score	
		Pearson *r*	*p* Value	Pearson *r*	*p* Value	Pearson *r*	*p* Value	Pearson *r*	*p* Value
DECT HU	L2	0.529	<0.01	0.524	<0.01	0.417	<0.01	0.408	<0.01
	Femur neck	0.351	<0.01	0.352	<0.01	0.593	<0.01	0.578	<0.01
	Distal femur	0.458	<0.01	0.450	<0.01	0.307	<0.01	0.286	<0.01
	Proximal tibia	0.342	<0.01	0.342	<0.01	0.286	<0.01	0.267	0.015
DECT BMD (g/cm^3^)	L2	0.585	<0.01	0.585	<0.01	0.476	<0.01	0.479	<0.01
	Femur neck	0.379	<0.01	0.384	<0.01	0.546	<0.01	0.550	<0.01
	Distal femur	0.458	<0.01	0.446	<0.01	0.454	<0.01	0.444	<0.01
	Proximal tibia	0.466	<0.01	0.479	<0.01	0.382	<0.01	0.381	<0.01

Pearson correlation analysis demonstrated significant correlation of dual-energy computed tomography (DECT) bone mineral density (BMD) and Hounsfield unit (HU) values with DXA (dual x-ray absorptiometry)-based BMD and T-score values.

**Table 3 medicina-58-01085-t003:** Diagnostic accuracy of DECT BMD and HU for osteoporosis diagnosis in distal femur and proximal tibia using DXA as the reference standard.

		L2 DXA as Standard of Reference						Femur Neck DXA as Standard of Reference					
						95% CI						95% CI	
		AUC	Cut-Off	Sensitivity	Specificity	Min	Max	AUC	Cut-Off	Sensitivity	Specificity	Min	Max
Distal femur	DECT BMD	0.872	89.2	82%	92%	0.809	0.936	0.714	96.9	70%	71%	0.616	0.813
	DECT HU	0.796	104.5	78%	77%	0.673	0.919	0.643	117.4	67%	65%	0.503	0.783
Proximal tibia	DECT BMD	0.935	78.3	85%	85%	0.871	0.999	0.738	80.9	76%	71%	0.609	0.866
	DECT HU	0.800	66.5	75%	77%	0.706	0.894	0.697	66.8	75%	71%	0.560	0.823

AUC: area under the curve. CI: confidence intervals. BMD: bone mineral density. HU: Hounsfield unit. DXA: dual-energy X-ray absorptiometry. Min: Minimum. Max: Maximum. L2: second lumbar spine.

## Data Availability

Data collected for this study, including individual patient data, will not be made available.

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
