# Peer review of "Dual-Energy CT-Based Bone Mineral Density Has Practical Value for Osteoporosis Screening around the Knee"

_medicina, 2022, doi:10.3390/medicina58081085_

Round 1

Reviewer 1 Report

Reviewer comments

Title Dual-energy CT-based Bone Mineral Density Has Practical Value for

I would like to thank the authors for their great work. However, Some important points need to be considered

·       English editing is needed in the whole manuscript

·       Title There are some missing words in the title

·       Abstract Handling abbreviations is not appropriate

The results should be informative enough to be understood

·       Materials and methods

Patients with metal prosthesis, fracture or infection in central

90 or peripheral regions were excluded…the authors did not determine regions of what

The authors mentioned the exclusion criteria from the start ..So ,it is not appropriate to exclude again fro the included patients ..this is not appropriate

The results of the study should be included at the results section not at the materials section

This section should be written in details and be informative about all methods used and how the data collected

·       Table 1

2 columns are present with no title at the top of each column to determine what they referred to?

The description of data is difficult to be understood ..Rewrite this table again in a more better mannar

·       Figures

The numbering of figures is not appropriate

What is the value of figure 2A

Figure 2A is not appropriate for publishing ..the writing is hazy and the authors should determine if its theirs or not

·       154 ..the results section will be more appropriate for thie paragraph  

·       Results

It is usually to start this section with some data about the sociodemographic data of the patients

·       Discussion

Add the strength and recommendations of the study

Author Response

  1. English editing is needed in the whole manuscript

Response: We understand the reviewer’s concern. However, actually, this manuscript has already gone through English editing by the charged copyedit compony, BioMed Proofreading LLC. We attached the certificate of English editing.

  1. Title There are some missing words in the title

Response: We understand  the reviewer’s comment. However, it is presumed that some technical error might be occurred.  We definitely have written the title as “Dual-energy CT-based Bone Mineral Density Has Practical Value for Osteoporosis Screening Around the Knee” in the original manuscript. Nonetheless, as suggested,  we updated the title into original version again. (Page 1 Line 2 – Page 1 Line 4)

Abstract

  1. Handling abbreviations is not appropriate

Response: We appreciate the reviewer’s comments. As suggested,  we  corrected  the missing abbreviations. (Page 1 Line 21 total knee arthroplasty (TKA), Page 1 Line 35 second lumbar vertebra (L2))

  1. the results should be informative enough to be understood

Response: We understand  the reviewer’s concern and agreed with comments. As suggested, we added more information about the diagnostic accuracy. (Page 1 Line 36- 37)

Materials and methods

  1. Patients with metal prosthesis, fracture or infection in central 90 or peripheral regions were excluded…the authors did not determine regions of what

Response: We appreciate and agreed with the reviewer’s comments. As suggested,  we  clarified the definite region. (Page 2 Line 93)

  1. The authors mentioned the exclusion criteria from the start ..So ,it is not appropriate to exclude again from the included patients ..this is not appropriate

Response: We understand the reviewer’s concern. However, we started from inclusion criteria first and it has been described the first sentence of Material and method section of the original manuscript (Line 86-89) Nonetheless, we rearranged the context. (Page 2 Line 94 – Page 3 Line 101)

  1. The results of the study should be included at the results section not at the materials section

Response: We understand  the reviewer’s concern. We   moved the corresponding sentences in the material and method section to the results section.. (Page 7 Line 188 – Line 196)

  1. This section should be written in details and be informative about all methods used and how the data collected

Response: We appreciate the reviewer’s suggestion.  As suggested, we described detailed method of confirming VOIs and ROIs in the materials and methods. And we also mentioned the software we used section (Page 6 Line 141 – Page 7 Line 165). The detailed variables are the values calculated by the software processing. Additionally, following researcher’s recommendation we added more detailed method of the assessment in Page 8 Line 168-169. We hope that these revisions address the reviewer’s concerns satisfactorily.  

  1. Table 1

2 columns are present with no title at the top of each column to determine what they referred to?

The description of data is difficult to be understood ..Rewrite this table again in a more better mannar

Response: As suggested, we corrected the confusing part of the table 1, 2 and 3.

  1. Figures

The numbering of figures is not appropriate

Response:. We arranged figure by number and alphabet like Figure 1, Figure 2A ~ 2C, Figure 3A ~ 3D and Figure 4A ~ 4B. And we corrected the wrong writing of Figure 2. C to Figure 2C in Page 6 Line 138 also.

  1. What is the value of figure 2A

Response: We appreciate the reviewer’s recommendation. Figure 2A is a result of assessment of dual x-ray absorptiometry (DXA) in the lumbar spine and femur neck. And the values are the T score and other variables taken from the DXA. Following the reviewer’s comment, to clarify the figure, we re-edited and simplified the figure 2A.

  1. Figure 2A is not appropriate for publishing ..the writing is hazy and the authors should determine if its theirs or not

Response: We appreciate the reviewer’s recommendation. Figure 2A is a result of assessment of dual x-ray absorptiometry (DXA) in the lumbar spine and femur neck. And the values are the T score and other variables taken from the DXA. As suggested, we corrected the Fig 2A  to clarify the figure (Page 6 Line 128)

Results

It is usually to start this section with some data about the sociodemographic data of the patients

Response: We appreciate the reviewer’s recommendation. As suggested, . we moved the corresponding sentence to the results section. (Page 8 Line 194 - 196)

Discussion

Add the strength and recommendations of the study

Response: We appreciate the reviewer’s recommendation. We added more detail about strength and possibility of expansion of this study. (Page 5 Line 356 ~ Line 365). And we mentioned the recommendation of this study in conclusion section. (Page 16 Line 394 ~ 397)

Reviewer 2 Report

The authors carried out the volumetric BMD using DECT and assessed its potential for osteoporosis prior to total knee arthroplasty. This manuscript adds, but not so substantially to the existing literature. I find it to be acceptable although there are a few comments that must be addressed. 

The title in the manuscript file is incomplete. 

The number of males and females are very different. I believe it reflects the fact that the number of arthroplasty done in females are much higher in females than males. The authors already acknowledged it in the discussion. But have the authors tried to differentiate the results for males and females? I expect them to be different because bone loss is significantly faster in females. 

Intraclass correlation coefficients were mentioned in the methods section but I do not see them in the results. 

Manual delineation of the trabecular VOI was only carried out by one of the authors. How did the authors ensure consistency? How many times was it repeated?

In addition to what was studied, It would be interesting to examine the correlation between the BMD and other parameters such as the BMI. 

Author Response

  1. The authors carried out the volumetric BMD using DECT and assessed its potential for osteoporosis prior to total knee arthroplasty. This manuscript adds, but not so substantially to the existing literature. I find it to be acceptable although there are a few comments that must be addressed. 

Response: We appreciate the reviewer’s understanding and succinct summary of our study.

  1. The title in the manuscript file is incomplete. 

Response: We understand the reviewer’s comment. However, it is presumed that some technical error might be occurred.  We definitely have written the title as “Dual-energy CT-based Bone Mineral Density Has Practical Value for Osteoporosis Screening Around the Knee” in the original manuscript. Nonetheless, as suggested,  we updated the title into original version again. (Page 1 Line 2 – Page 1 Line 4)

  1. The number of males and females are very different. I believe it reflects the fact that the number of arthroplasties done in females are much higher in females than males. The authors already acknowledged it in the discussion. But have the authors tried to differentiate the results for males and females? I expect them to be different because bone loss is significantly faster in females. 

Response: We appreciate the reviewer’s comment and recommendation. Unfortunately, we didn’t compare the result between sex. Because as we mentioned and you’ve already know, although it is not yet clear why the majority of patients with arthritis in Korea are females, the majority patient with arthritis in South Korea are females. And the number of male patients is to small. Although, we didn’t inspecd the possible factors affecting BMD in peripheral regions like sex, BMI, comorbidities etc in this study, if this analysis is done it would give more valuable information. Thanks to your kind recommendation, we would consider that kind of analysis in further study. We agreed on your recommendation, and we added these comment in limitation part of discussion section. (Page 16 Line 386 ~ Line 389)

  1. Intraclass correlation coefficients were mentioned in the methods section, but I do not see them in the results. 

Response: We appreciate the reviewer’s comment. We’ve mentioned the result of intra and inter observer reliability for all radiographic measurements in method section as “Intra- and inter-observer reliability for all radiographic measurements was considered acceptable, ranging from 0.81 to 0.99 and 0.81 to 0.96, respectively.” We move the result of intra- and inter- observer reliability to the end of the result section (Page 10 Line 231 – Page 10 Line 233).

  1. Manual delineation of the trabecular VOI was only carried out by one of the authors. How did the authors ensure consistency? How many times was it repeated?

Response: We understand the reviewer’s comment. However, measurement of the radiologic findings including the trabecular VOI were done by two times by two different authors respectively. We’ve already described it in the materials and method section. To clarify, we added a sentence that  “Each researcher assessed every radiologic variables two time with the interval at least two weeks” to  the materials and methods section. (Page 8 Line 168 – Line 169)

  1. In addition to what was studied, It would be interesting to examine the correlation between the BMD and other parameters such as the BMI. 

Response: We appreciate the reviewer’s recommendation. We appreciate the reviewer’s comment and recommendation. Unfortunately, we didn’t inspected the possible factors affecting BMD in peripheral regions like sex, BMI, comorbidities etc in this study. If this analysis is done it would give more valuable information. Thanks to your kind recommendation, we would consider that kind of analysis in further study. We would consider analyzing the correlation between BMD of DXA and DECT with other parameters such as BMI in the further study. We agreed on your recommendation, and we added these comment in limitation part of discussion section. (Page 16 Line 386 ~ Line 389)

Round 2

Reviewer 1 Report

Accept